# Divergence of Neural Tangent Kernel in Classification Problems

**Zixiong Yu**[*]
Noah's Ark Lab
Huawei Technologies Ltd.
Shenzhen, Guangdong, China
`yuzixiong2@huawei.com`

**Songtao Tian**[†]
Department of Mathematical Sciences
Tsinghua University
Haidian District, Beijing, China
`tiansongtao.2020@tsinghua.org.cn`

**Guhan Chen**[‡]
Department of Statistics and Data Science
Tsinghua University
Haidian District, Beijing, China
`chen-gh23@mails.tsinghua.edu.cn`

## Abstract

This paper primarily investigates the convergence of the Neural Tangent Kernel (NTK) in classification problems. This study firstly show the strictly positive definiteness of NTK of multi-layer fully connected neural networks and residual neural networks. Then, through a contradiction argument, it indicates that, during training with the cross-entropy loss function, the neural network parameters diverge due to the strictly positive definiteness of the NTK. Consequently, the empirical NTK does not consistently converge but instead diverges as time approaches infinity. This finding implies that NTK theory is not applicable in this context, highlighting significant theoretical implications for the study of neural networks in classification problems. These results can also be easily generalized to other network structures, provided that the NTK is strictly positive definite.

## 1 Introduction

Recently, neural network have achieved remarkable performance in various fields, such as computer vision, natural language processing, and generative models. In the field of computer vision, Krizhevsky et al. (2012) proposed AlexNet, which significantly outperformed traditional methods in the ImageNet competition using deep convolutional network. Subsequently, He et al. (2016) introduced ResNet, which addressed the degradation problem in training deep network by incorporating residual blocks, significantly improving model performance. In the field of natural language processing, Vaswani et al. (2017) proposed the Transformer, which greatly enhanced sequence-to-sequence tasks through the self-attention mechanism. In the field of generative models, Kingma & Welling (2013) introduced the Variational Autoencoder (VAE), enabling efficient training of generative models. However, despite the excellent performance of network in practical applications, there is still no complete theoretical explanation for why network perform so well.

The theoretical properties of network have been a subject of study for a long time. In terms of the expressive power of network, Hornik et al. (1989); Cybenko (1989) proposed the well-known universal approximation theorem, which demonstrates that a neural network with a sufficient number of parameters can approximate any continuous function. Recently, many works have continued to explore the expressive power of network in broader and more structured contexts, such as (Cohen et al., 2016; Hanin & Sellke, 2017; Lu et al., 2017), among others. However, these works focus solely on the model's expressive capabilities of network and neglect their statistical properties, thus failing to fully explain the excellent performance of network.

---

[*]First author. [†]Co-first author. [‡]Corresponding author.

Regarding the statistical properties of network, many studies have focused on the generalization ability of network. Within the static non-parametric regression framework, Bauer & Kohler (2019); Schmidt-Hieber (2020) investigated the minimax optimality of network when different Hölder function classes are used as the true target functions. At the same time, the dynamic training of network is equally important. Du et al. (2018); Allen-Zhu et al. (2019); Chizat et al. (2019) studied the performance of sufficiently wide network on the training set after gradient descent and stochastic gradient descent in regression problems. Building on this, Jacot et al. (2018) explicitly proposed the concept of the Neural Tangent Kernel (NTK) to more precisely characterize the performance of sufficiently wide network during gradient descent.

In the NTK theory framework, research can be typically divided into two steps: studying the convergence of the NTK, i.e., whether the neural network kernel can converge to the NTK; and investigating the performance of the corresponding NTK regressor within the kernel regression framework. The former establishes the validity of the NTK theory, while the latter elucidates the properties of network. For instance, Arora et al. (2019); Lee et al. (2019); Hu et al. (2021) verified the NTK convergence of various networks under the MSE loss function. Furthermore, Lai et al. (2023a); Li et al. (2023) demonstrated the uniform convergence of the NTK for fully-connected networks directly trained with MSE. Additionally, many work have investigated the generalization ability of the NTK regressor in MSE regression problem (Lee et al., 2019; Hu et al., 2021; Suh et al., 2021; Lai et al., 2023a; Li et al., 2023; Tian & Yu, 2024). However, for networks trained using the cross-entropy function, which is common in classification problems, the property of the empirical NTK is still an open question.

This article primarily investigates the convergence of NTK in classification problem. As mentioned above, the convergence of NTK has been thoroughly studied in regression problem with the MSE loss function. In classification problems, the cross-entropy function is more suitable for handling categorical label data, and thus it has a wider range of applications compared to MSE. The convergence of NTK during training with the cross-entropy loss function, which is necessary, however, has hardly been researched. According to our results, under the cross-entropy loss function, the NTK of multi-layer fully connected neural network and residual neural network will no longer uniformly converge to NTK but instead diverge as time approaches infinity. This implies that NTK theory is no longer applicable in this case. From this perspective, our work has significant theoretical implications.

## 1.1 RELATED WORKS

The concept of NTK was first introduced by Jacot et al. (2018) to approximate the training process of wide network using kernel regression. There has been substantial research on the convergence of NTK in regression problems, with early works focusing on the pointwise convergence of Neural Network Kernels (NNK) to NTK (Du et al., 2019; Allen-Zhu et al., 2019; Arora et al., 2019). Building on this, many have proven the uniform convergence of NTK, meaning that as the network width approaches infinity, NNK uniformly converges to NTK across all possible samples and at every point in the training process (Lai et al., 2023a; Li et al., 2023; Lai et al., 2023b; Chen et al., 2024). For NTK in classification problems, due to the complex nature of the cross-entropy loss function, many researchers have used MSE loss as a substitute (Vyas et al., 2022; Hron et al., 2020). Some studies also examine the generalization ability of neural networks for classification problems under the cross-entropy loss function from different perspectives. Recent works, such as Ji & Telgarsky (2019); Taheri & Thrampoulidis (2024), have explored the dynamics of neural networks under the regime where the network width $m \to \infty$ and the training time $T$ is fixed and finite. These studies demonstrate that under such conditions, the NTK regime holds during training. In contrast, our work examines the case where $m$ is fixed and $T \to \infty$. We show that in this setting, the NTK exhibits divergence, preventing uniform convergence to kernel predictors and revealing distinct overfitting dynamics in classification tasks.

## 1.2 CONTRIBUTION

In this article, we study the convergence of NTK in classification problems for fully connected neural network and residual neural network. We point out that when using the cross-entropy loss function commonly used in classification problems, the NTK of network cannot achieve uniform convergence over time and across all training samples.

First, we investigate the strict positive definiteness of NTK for fully connected network and residual network. We consider network defined on a general compact domain, i.e., input samples $x \in \mathcal{X} \subset \mathbb{R}^d$. Through appropriate transformations, we convert the NTK of fully connected network and residual network into a recursive form and an arc-cosine kernel form, respectively, and then prove their strict positive definiteness. After proving the strict positive definiteness of NTK, we analyze the dynamic properties of network and use a proof by contradiction to demonstrate that under the cross-entropy loss function, the NTK of network cannot achieve uniform convergence.

To the best of our knowledge, our article is the first to theoretically prove the divergent property of NTK in classification problems. This aligns with the experiment results and reflects the limitations of earlier theoretical work on related issues. Overall, our article demonstrates that using NTK to study the properties of network in classification problems is not a good choice. Our results can also be easily generalized to other network structures, provided that the NTK is strictly positive definite.

## 2 PRELIMINARIES

### 2.1 NOTATIONS

For two sequence $\{x_n\}$ and $\{y_n\}$, we denote by $x_n = O(y_n)$ if there exists some positive constant $C$ such that $|x_n| \leq C|y_n|$ holds for any large enough $n$. For index sequence $i = 1, \cdots, n$, we write as $i \in [n]$ for notational simplicity. We denote the sample marix as $X = (x_1^T, x_2^T, \cdots, x_n^T)^T$, and $Y = (y_1, y_2, \cdots, y_n)^T$. For a function $f(\cdot) : \mathbb{R}^d \to \mathbb{R}$, we denote $f(X)$ to represent the entry-wise application of the function to each element in $X$. That is, $f(X) = (f(x_1), f(x_2), \cdots, f(x_n))^T$. This means the function $f$ is applied individually to each sample in the sample matrix. We denote by $[n]$ the sequence $1, 2, \cdots, n$. In this paper, we consider the binary classification problem. We assume the pairs of training data $(x, y)$ comes from $\mathcal{X} \times \{0, 1\}$, where $\mathcal{X}$ is a compact subset of $\mathbb{R}^d$. The label $y_i$ comes from $\{0, 1\}$ with unknown distribution. It is worth noting that, we do not make requirements on the separability of data, as well as other characteristics.

### 2.2 NEURAL NETWORK

**Fully connected network** As a simple but representative kind of network, we consider fully connected neural network with multi-hidden layer:

$$
\begin{aligned}
&\alpha^{(0)}(x) = x; \\
&\alpha^{(l)}(x) = \sqrt{\frac{2}{m_l}} \sigma \left( W^{(l)} \alpha^{(l-1)}(x) + b^{(l)} \right); \quad , l = 1, 2, \cdots, L \\
&f(x; \theta) = W^{(L+1)} \alpha^{(L)}(x)
\end{aligned}
\tag{2.1}
$$

**Residual network** The residual network is also a kind of widely used network, which features skip connections as following.

$$
\begin{aligned}
&\alpha^{(0)}(x) = \sqrt{\frac{1}{m_0}} (Ax + b); \\
&\widetilde{\alpha}^{(l)}(x) = \sqrt{\frac{2}{m_l}} \sigma \left( W^{(l)} \alpha^{(l-1)}(x) + b^{(l)} \right), \quad l = 1, 2, \cdots, L; \\
&\alpha^{(l)}(x) = \alpha^{(l-1)}(x) + \alpha \sqrt{\frac{1}{m_l}} \left( V^{(l)} \widetilde{\alpha}^{(l)}(x) + d^{(l)} \right), \quad l = 1, 2, \cdots, L; \\
&f(x; \theta) = W^{(L+1)} \alpha^{(L)}(x).
\end{aligned}
\tag{2.2}
$$

In both two models, $\sigma$ is the entry-wise ReLU function as $\sigma(x) := \max(0, x)$, and we use $\theta$ to denote the vector that the parameters above flatten to be. For $l \in [L]$, the dimension of matrices $W^{(l)}$ and $b^{(l)}$ are $\mathbb{R}^{m_l \times m_{l-1}}$ and $\mathbb{R}^{m_l \times 1}$, respectively. Additionally, the dimension of $W^{(L+1)}$ is $\mathbb{R}^{1 \times m_L}$. Since residual networks introduce skip connections compared to fully connected networks (the third equation in eq. (2.2)), the width of each layer in a residual network is the same, that is, $m_0 = m_1 = \cdots = m_L$, where weights $V^{(l)} \in \mathbb{R}^{m_l \times m_l}$ and bias $d^{(l)} \in \mathbb{R}^{m_l}$. The parameter $\alpha$ in

eq. (2.2) is a chosen scaling factor. Additionally, since the input dimension and width may also differ, an additional linear transformation is applied at the input layer (see the first equation in eq. (2.2)), with weights $A \in \mathbb{R}^{m_0 \times d}$ and bias $b \in \mathbb{R}^{m_0}$. At initialization, for the fully connected neural network, the parameters of $W^{(l)}$ and $b^{(l)}$ are independently and identically distributed (i.i.d.) as standard Gaussian variables $\mathcal{N}(0, 1)$. For the residual network, all parameters are initialized as i.i.d. random variables from $\mathcal{N}(0, 1)$, except for $b^{(l)}$ and $d^{(l)}$, which are initialized to zero. This exception is made solely for the convenience of NTK calculation and is not the main focus of this paper.

**Loss function**    For classification problem, we choose to apply the commonly-used and representative Cross-Entropy loss function, although our proof can be generalized to a class of loss function as mentioned in Remark 1. That is

$$
\begin{aligned}
\mathcal{L}(\theta) &= -\sum_{i=1}^{n} [y_i \ln(o_i) + (1 - y_i) \ln(1 - o_i)] \\
&= \sum_{i=1}^{n} \ell\left((2y_i - 1)f(x_i; \theta)\right),
\end{aligned}
\tag{2.3}
$$

where $\{o_i\}_{i=1}^{n}$ and function $\ell$ is defined by

$$
o_i := \frac{1}{1 + e^{-f(x_i; \theta)}}; \quad \ell(x) := \ln(1 + e^{-x}).
\tag{2.4}
$$

Under such setting, $o_i$ can be regarded as the "output probability", since it transform $f(x_i; \theta)$ and then compare to the label $y_i$. In this way, the network function $f(x; \theta)$ is expected to be positive correlated to the probability that the label $y$ of $x$ to be label 1 instead of 0. We use $u_i$ to denote:

$$
u_i := |o_i - y_i|.
\tag{2.5}
$$

Then we have

$$
u_i = \frac{1}{1 + e^{(2y_i - 1)f(x_i; \theta)}} = \begin{cases} o_i & \text{if } y_i = 0; \\ 1 - o_i & \text{if } y_i = 1. \end{cases}
\tag{2.6}
$$

We can see $u_i \in [0, 1]$. Under the Cross Entropy loss function, we use gradient flow to approximate the gradient descent optimization process. There is an simple way to represent the gradient by $u_i$. The dynamic euqation is :

$$
\begin{aligned}
\frac{\mathrm{d}}{\mathrm{d}t} \theta_t &= -\nabla_\theta \mathcal{L}(\theta_t) = -\sum_{i=1}^{n} \ell'((2y_i - 1)f(x_i; \theta_t))(2y_i - 1)\nabla_\theta f(x_i; \theta_t) \\
&= \sum_{i=1}^{n} \nabla_\theta f(x_i; \theta_t)(2y_i - 1)u_i.
\end{aligned}
\tag{2.7}
$$

## 3    NEURAL TANGENT KERNEL

We first introduce the concepts of NNK and NTK. Both can be used to study the dynamics of neural networks during training and have therefore been widely discussed in previous literature.

**Neural Network Kernel**    The Neural Network kernel (NNK), also called as the empirical NTK, is the inner product of the derivatives of the neural network with respect to all parameters. We denote NNK by $K_t(\cdot, \cdot) : \mathbb{R}^d \times \mathbb{R}^d \rightarrow \mathbb{R}$ . Specifically, the expression is given by

$$
K_t(x, x') = \langle \nabla_\theta f(x; \theta_t), \nabla_\theta f(x'; \theta_t) \rangle.
\tag{3.1}
$$

In equation 2.7, we know the dynamic of parameters during training process. Since $f(x; \theta_t)$ is a function that decided by $\theta_t$, its dynamic can also be derived by:

$$
\begin{aligned}
\frac{\mathrm{d}}{\mathrm{d}t} f(x; \theta_t) &= [\nabla_\theta f(x; \theta_t)]^T \left[ \frac{\partial \theta_t}{\partial t} \right] \\
&= \sum_{i=1}^{n} [\nabla_\theta f(x; \theta_t)]^T [\nabla_\theta f(x_i; \theta_t)] (2y_i - 1)u_i. \\
&= \sum_{i=1}^{n} K_t(x, x_i)(2y_i - 1)u_i,
\end{aligned}
\tag{3.2}
$$

where $K_t(x, x_i) = [\nabla_\theta f(x; \theta_t)]^T [\nabla_\theta f(x_i; \theta_t)]$. Note that equation 3.2 has similar form to kernel logistic regression. Unlike traditional kernel regression, the kernel function in 3.2 is not fixed during training. Instead, it changed over time.

**Neural Tangent kernel**    To better understand the dynamic characteristic of network output function, many previous work (Jacot et al., 2018; Allen-Zhu et al., 2019; Arora et al., 2019) systematically studied the properties of NNK, and showed that as the network width $m$ comes to infinity, NNK will converges to a fixed kernel both at initialization and during the whole training process, which is the so-called Neural Tangent Kernel (NTK):

$$K_t(x, x') \xrightarrow{\mathbb{P}} K(x, x'). \tag{3.3}$$

Moreover, Lai et al. (2023a); Li et al. (2023) further shows that the convergence from NNK to NTK is uniform over all possible input vector and all time point. This phenomenon helps a lot in the learning of generalization ability network under the frame work of NTK theory.

In regression problems, the convergence of NNK is built on the convexity of the MSE loss function. However, in classification problems where Cross Entropy is commonly used as the loss function, we can show that this convergence no longer holds. We will elaborate on this point in the subsequent sections, which also reflects the limitations of NTK theory.

### 3.1    STRICTLY POSITIVE DEFINITENESS OF NTK

In order to ensure the divergence relationship between NNK and NTK, strictly positive definiteness of the kernel function is crucial. We first explicitly recall the following definition of positive definiteness to avoid potential confusion.

**Definition 1** (Strictly positive definiteness). A kernel function $K(\cdot, \cdot) : \mathcal{X} \times \mathcal{X} \to \mathbb{R}$ is strictly positive definite over domain $\mathcal{X}$ if for any positive integer $n$ and any $n$ different points $x_1, \ldots, x_n \in \mathcal{X}$, the smallest eigenvalue $\lambda_{\min}$ of the matrix $K(X, X) = (K(x_i, x_j))_{1 \le i, j \le n}$ is positive.

To derive the positive definiteness of NTK of fully connected network and residual network, we first demonstrate the their expression in the recursive form or explicit form, respectively.

**NTK of Fully connected network**    We first present the recursive formula of NTK of fully connected network. For fully connected network given by equation 2.1, the NTK denoted by $K^{\mathrm{FC}}$ can be defined as following as shown in Jacot et al. (2018). We first define

$$\Theta^{(1)}(x, x') = \Sigma^{(1)}(x, x') = \langle x, x' \rangle + 1, \tag{3.4}$$

and

$$\Theta^{(l+1)}(x, x') = \Theta^{(l)}(x, x')\dot{\Sigma}^{(l+1)}(x, x') + \Sigma^{(l+1)}(x, x'), \tag{3.5}$$

where

$$\begin{aligned}
\Sigma^{(l+1)}(x, x') &= 2\left(\mathbf{E}_{f \sim \Sigma^{(l)}}[\sigma(f(x))\sigma(f(x'))] + 1\right), \\
\dot{\Sigma}^{(l+1)}(x, x') &= 2\mathbf{E}_{f \sim \Sigma^{(l)}}[\dot{\sigma}(f(x))\dot{\sigma}(f(x'))].
\end{aligned} \tag{3.6}$$

for any $l \in 1, \cdots, L$. Then the formula of $K^{\mathrm{FC}}$ is

$$K^{\mathrm{FC}}(x, x') = \Theta^{(L+1)}(x, x'). \tag{3.7}$$

**NTK of Residual network**    We can provide the NTK expression for residual neural network 2.2, based on the the specific expression of NTK under the network structure without bias term which is given by Huang et al. (2020). Let $x, x' \in \mathbb{R}^d$ be two samples. Introduce the following functions:

$$\kappa_0(u) = \frac{1}{\pi}(\pi - \arccos u), \quad \kappa_1(u) = \frac{1}{\pi}\left(u(\pi - \arccos u) + \sqrt{1 - u^2}\right),$$

and

$$\widetilde{x} = (x, 1), \quad \phi(x) = \frac{\widetilde{x}}{\|\widetilde{x}\|_2}, . \tag{3.8}$$

The NTK of $L$ hidden layers ResNet, denoted as $K^{\text{Res}}$ is given by

$$K^{\text{Res}}(x,x') = \|\widetilde{x}\| \left\|\widetilde{x'}\right\| \left[ r^{(L)}(\phi(x),\phi(x')) + [\phi(x)]^T \phi(x') B_1(\phi(x),\phi(x')) + K_L(\phi(x),\phi(x')) \right] \tag{3.9}$$

where $r^{(L)}(x,x')$ (Huang et al., 2020) is defined as

$$r^{(L)}(x,x') = C_L \sum_{l=1}^{L} B_{l+1} \left[ (1+\alpha^2)^{l-1} \kappa_1 \left( \frac{K_{l-1}}{(1+\alpha^2)^{l-1}} \right) + (K_{l-1} + 1) \cdot \kappa_0 \left( \frac{K_{l-1}}{(1+\alpha^2)^{l-1}} \right) + 1 \right] \tag{3.10}$$

where

$$K_l = K_{l-1} + \alpha^2(1+\alpha^2)^{l-1} \kappa_1 \left( \frac{K_{l-1}}{(1+\alpha^2)^{l-1}} \right); \quad K_0 = x^\top x';$$

$$B_l = B_{l+1} \left[ 1 + \alpha^2 \kappa_0 \left( \frac{K_{l-1}}{(1+\alpha^2)^{l-1}} \right) \right]; \quad B_{L+1} = 1, \quad C_L = \alpha^2$$

for $l \in [L]$. In the above equations, $K_l$ and $B_l$ are abbreviations for $K_l(x,x')$ and $B_l(x,x')$, respectively.

For the NTK defined above, we have the following proposition on their positive definiteness, which is proved in Appendix:

**Proposition 1** (Strictly positive definiteness of NTK). *The NTK of fully connected network 2.1 and residual network 2.2 is strictly positive definite.*

The strictly positive definiteness has long been a topic of interest in NTK theory (Jacot et al., 2018; Zhang et al., 2024; Nguyen et al., 2021). The strictly positive definiteness of NTK of fully connected network defined on the unit sphere $\mathbb{S}^{d-1}$ was first proved by Jacot et al. (2018). Recently, Lai et al. (2023a) proved the strictly positive definiteness of NTK for one-hidden-layer biased fully connected neural networks on $\mathbb{R}$, and Li et al. (2023) generalized it to multiple layer fully connected NTK on $\mathbb{R}^d$ ($d \geq 1$). In these works, the bias term $b^{(l)}$ is omitted as a simplification of the network structure. In this work, we adopt a commonly used fully connected neural network with bias terms and use a recursive form to prove strictly positive definiteness. As to the NTK of residual network, we use equation 3.9 to transform the input vector from $\mathbb{R}^d$ to $\mathbb{S}^d_+$, and then prove the strictly positive definiteness of the dot-product kernel $r^{(L)}$.

## 4 MAIN RESULTS

### 4.1 DIVERGENCE OF NETWORK

At standard NTK initialization, the NNK will converges in probability as width comes to infinity (Arora et al., 2019), which is evidently independent of the loss function and training method. In regression problems, NTK is also proved to be convergent during the training process, which forms the theoretical basis for studying the generalization ability of neural networks in NTK theory (Suh et al., 2021; Li et al., 2023; Lai et al., 2023a). This phenomenon is because the parameters will not deviate far from their initial values no matter how long the network is trained, which is known as the so-called NTK regime (Allen-Zhu et al., 2019). However, it no longer holds when training with the cross-entropy loss function. We denote by $\widetilde{\lambda}_0(t)$ as the minimum eigenvalue of the NNK matrix: $\widetilde{\lambda}_0(t) = \lambda_{\min}(K_t(X,X))$.

**Theorem 1.** *Fix the training samples $\{(x_i, y_i)\}_{i \in [n]}$. Consider fully-connected network and residual network with cross entropy loss function in classification problem. If $\widetilde{\lambda}_0(t)$ is consistently lower bounded by some postive constant $C$ during training, then the network output function will comes to infinity at the sample points $\{x_i\}_{i \in [n]}$, i.e.,*

$$\lim_{t \to \infty} |f_t(x_i)| = +\infty. \tag{4.1}$$

This theorem reflects the divergence property of network in classification problems. We attempt to understand the theorem above from a less technical perspective and in a simple, easy-to-understand manner. If we have that the NNK always maintain a uniformly strictly positive definiteness, i.e.,

$\widetilde{\lambda}_0(t) \geq C > 0$, then at each input vector $x_i$ in the sample set, the output function value of network will tend to infinity, which means that some parameters of network diverge during the training process as time changes. We demonstrate the following direct corollary of Theorem 1:

**Corollary 1** (Failure of the NTK regime). *Assume the conditions in Theorem 1 holds. For any initialized parameter $\theta_0$, after training we have that*

$$\limsup_{t \to \infty} \|\theta_t - \theta_0\|_\infty = \infty \tag{4.2}$$

Note that in Theorem 1, we do not impose any requirements on the width of network, meaning that the aforementioned divergence holds regardless of the width $m$. Of course, we have not forgotten that all of this is based on the condition $\widetilde{\lambda}_0(t) \geq C > 0$. This condition will be satisfied if NNK uniformly converges to NTK, which will be used in the proof by contradiction of Theorem 2.

## 4.2 DIVERGENCE OF NTK

In this section, we will discuss the divergence of NTK in classification problem and demonstrate our main results. In infinite time, the NTK continues to evolve and will not converges to a fixed NTK like in the regression problem case. Now we demonstrate our main theorem.

**Theorem 2** (Divergence of NTK of fully connected network and residual network ). *Given samples $\{(x_i, y_i)\}_{i=1}^n$ and consider the training process under cross-entropy function, we have*

- *Let $\lambda_0 := \lambda_{\min}(K^{\mathrm{FC}}(X, X))$. For any initialized parameter $\theta_0$, there exists $x, x' \in \mathcal{X}$, such that*

$$\sup_{t \geq 0} |K_t^{\mathrm{FC}}(x, x') - K^{\mathrm{FC}}(x, x')| \geq \frac{\lambda_0}{2n^2}. \tag{4.3}$$

- *Let $\lambda_0 := \lambda_{\min}(K^{\mathrm{Res}}(X, X))$. For any initialized parameter $\theta_0$, there exists $x, x' \in \mathcal{X}$, such that*

$$\sup_{t \geq 0} |K_t^{\mathrm{Res}}(x, x') - K^{\mathrm{Res}}(x, x')| \geq \frac{\lambda_0}{2n^2}. \tag{4.4}$$

To see the significance of this result, we can compare it with regression problems: In regression, regardless of the value of $n$, as long as $m$ is sufficiently large, we can achieve uniform convergence of NNK to NTK, then approximate neural networks using kernel regression, and thus obtain properties of generalization ability (Lai et al., 2023a; Li et al., 2024). However, in classification problems, NNK cannot uniformly converge to NTK. Although when $n$ is sufficiently large, the lower bound of the gap between NNK and NTK, $\frac{\lambda_0}{2n^2}$, is small, this small lower bound still makes it difficult for us to approximate neural networks using kernel regression. The theorem indicates that for both fully-connected and residual network, the NTK does not remain fixed during training when using the cross-entropy loss function. Instead, it continues to evolve, suggesting that the NTK theory, which works well for regression problems, does not directly apply to classification problems. This divergence implies that the NTK in classification tasks cannot provide a static approximation of the training dynamics, thereby necessitating new approaches to understand and analyze the behavior of neural network.

## 5 PROOF SKETCH

We present the sketch of our proofs in this part and defer the complete proof to Appendix. In this section, we denote the network output function as $f^{\mathrm{NN}} : \mathcal{X} \subset \mathbb{R}^d \to \mathbb{R}$ and denote the corresponding NNK and NTK as $K_t^m(\cdot, \cdot)$ and $K(\cdot, \cdot)$, respectively. We also fix samples $\{(x_i, y_i)\}_{i \in [n]}$. To simplify the notation, in the proof sketch we do not deliberately distinguish the symbols for different network structures. Namely, the above symbols represent both fully connected neural networks and residual neural networks. We assume the training samples of the neural network are trained using the cross-entropy loss function.

We can finish the proof of Theorem 2 by contradiction. For the sake of contradiction, we firstly assume that during training, NNK is invariant:

$$\sup_{x, x' \in \mathcal{X}} \sup_{t \geq 0} |K_t^m(x, x') - K(x, x')| = o_m(1). \tag{5.1}$$

This implies that we also have the relationship between the minimum eigenvalues

$$\sup_{t\geq 0}|\lambda_{\min}\left(K_t^m(X,X)\right) - \lambda_{\min}\left(K(X,X)\right)| \leq \sup_{t\geq 0}\|K_t^m(X,X) - K(X,X)\|_{\mathrm{F}} = o_m(1). \quad (5.2)$$

Recall that the strictly positiveness of NTK has been demonstrated in Section 3.1, we know NNK is also consistently strictly positive definite during training. Namely, there exists a positive constant $C$ such that

$$\inf_{t\geq 0}\lambda_{\min}(K_t^m(X,X)) \geq C. \quad (5.3)$$

Since NNK involves the derivatives of the network with respect to the parameters, it can be used to analyze the dynamics of the network function during training. In equation 2.5 we define $u \in \mathbb{R}^n$ , which represents the residual and also reflects the output value of the network function on the training set. We construct

$$V(u) := \sum_{i=1}^{n} \ln(1 - u_i(t)), \quad \text{and} \quad V_t := V(u(t)) \quad (5.4)$$

For the scalar $V_t$, we have the dynamical equation

$$\frac{\mathrm{d}}{\mathrm{d}t}V_t = -u^T(t)\mathrm{diag}(2Y - 1)K_t^m(X,X)\mathrm{diag}(2Y - 1)u(t). \quad (5.5)$$

Since NNK is strictly positive definite in equation 5.3, we know that $V_t$ is monotonically decreasing. Through some technical analysis (details in the appendix), we finally get $V_t \rightarrow 0$ as $t \rightarrow \infty$. Therefore, as $t \rightarrow \infty$, we have

$$u(t) \rightarrow 0, \quad \text{and} \quad |f_t^{\mathrm{NN}}(x_i)| \rightarrow \infty. \quad (5.6)$$

Combined with the specific network structure, we can derive that

$$\lim_{t\rightarrow\infty}\sup_{x,x'\in\{x_i\}_{i=1}^n}|K_t^m(x,x')| \rightarrow \infty, \quad (5.7)$$

which indicates that NNK diverges during training. Since the assumption of uniform convergence of NNK to NTK leads to NTK being bounded, this results in a contradiction, and we finish the proof.

## 6 NUMERICAL EXPERIMENTS

### 6.1 SYNTHETIC DATA

We conducted experiments on a synthetic dataset, using the previously mentioned fully connected network and residual network architectures.

**Divergence of the Fully Connected Network Function**   The fully connected network has three hidden layers, with an input dimension of $d = 2$ and a width of $m = 2000$. For the training set, we generate 6 input vectors uniformly distributed on the unit sphere $\mathbb{S}^d$, i.e., the points $\{x_i\}_{i\in[6]}$ are $(\cos\theta_i, \sin\theta_i)$, where $\theta_i = \frac{i\pi}{3}$. The labels of the points are $(0, 1, 0, 1, 0, 1)$, respectively. We shuffle the labels to eliminate any potential influence of data separability. The learning rate is set to $0.1$, and the network is trained for $10,000$ epochs. The output values of the network at the 6 training points are plotted during training, and the final result is shown in Figure 1.

**Divergence of the NNK during training**   In this section, we examine the behavior of the NTK both at initialization and during training to highlight the impact of the cross-entropy loss function. At initialization, we plot the NTK function of network with different width by fixing $x = (1, 0)$ and varying the polar angle of $x'$ from $-\pi$ to $\pi$. The result is shown in in Figure 2, which demonstrates that the NTK converges at $t = 0$, particularly when $m = 2000$. The results show that the NNK approximately converges to the NTK, aligning with previous findings in NTK theory. We then train the network with a width of $m = 2000$ using the cross-entropy loss function and plot the network's output at the same points. Figure 3 presents the behavior of the NTK across different epochs, highlighting its divergence during the training process.

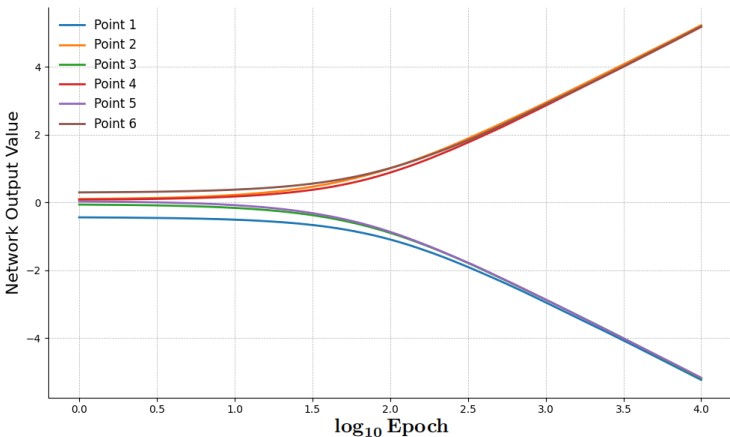

Figure 1: Divergence of the fully connected network: The plot shows the output values of the network at the 6 training points over the training process. Despite starting from a shuffled label configuration, the network function diverges at all six points as training progresses.

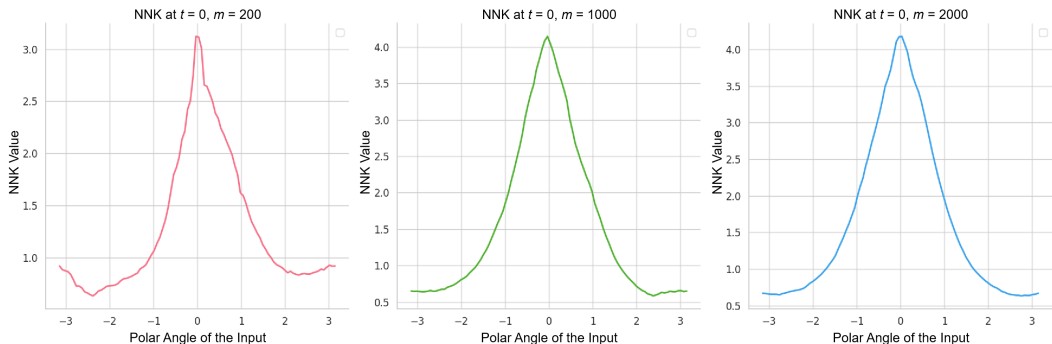

Figure 2: Convergence of the fully connected NTK: The NTK functions at initialization for networks with different widths ($m = 200$, $m = 1000$, and $m = 2000$). The results show convergence of the NTK at $t = 0$, with the convergence becoming more apparent as the network width increases, especially when $m = 2000$.

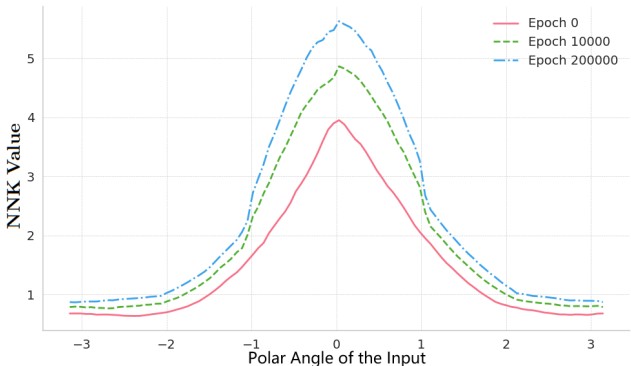

Figure 3: Divergence of the fully connected NTK: The behavior of the NTK at different epochs during training for a network with width $m = 2000$. The plot highlights the divergence of NTK over the course of training.

## 6.2 REAL DATA

We conducted an experiment on the MNIST dataset, using parity (odd or even) as the criterion for binary classification. We trained a four-layer fully connected neural network on the MNIST dataset. The network has a width of $m = 500$, with a learning rate of lr $= 0.5$, and was trained for epoch $= 100,000$. Since the dimension of MNIST dataset is $d = 784$, it is hard to plot the value of NTK like in Figure 4. Therefore, we pick three points and show that the empirical NTK value disconverges on these points, which is sufficient to confirm our results. The result is shown in Figure 4. The values of the first 10 epochs are discarded here in order to focus more on the behavior during training and avoid the experimental results being affected by initialization.

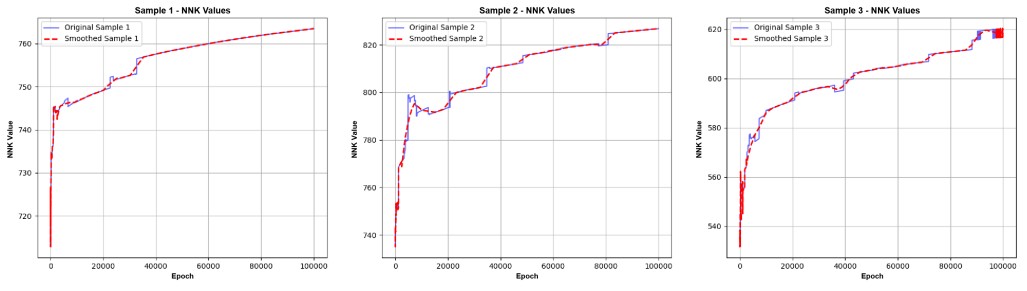

Figure 4: The NTK values on the MNIST dataset for three selected points. The blue lines represent the original NTK values, while the red dashed lines show the smoothed values. The NTK values were computed over $100,000$ epochs for a four-layer fully connected neural network with a width of $m = 500$ and a learning rate of lr $= 0.5$. These points were chosen to illustrate the disconvergence of the empirical NTK values.

## 7 CONCLUSION

In this paper, we investigated the behavior of NTK of fully connected network and residual network in classification problems, particularly focusing on the convergence properties when training with the cross-entropy loss function. Our study reveals that unlike in regression problems, where NTK convergence has been well-documented, NTK in classification tasks does not uniformly converge and instead diverges as training progresses. This divergence indicates a limitation in applying NTK theory to classification problems and suggests that NTK does not provide a static approximation of the training dynamics in these scenarios.

Our analysis highlights the need for new theoretical tools and frameworks to better understand and analyze neural network in the context of classification. We have shown that the standard NTK regime fails in classification problems, which implies that current theories might not fully explain the generalization properties of network trained on such tasks. This opens up new avenues for research in understanding the complex dynamics of neural network beyond the NTK framework, especially when dealing with classification problems where cross-entropy loss is prevalent.

### ACKNOWLEDGMENTS

All three authors of this paper have either participated in discussions at or are currently studying in the Department of Statistics and Data Science at Tsinghua University. We would like to express our sincere gratitude to the institution, our advisor, and fellow students for their guidance, training, and valuable exchanges. We also extend our thanks to the AI Data Technology Laboratory of Huawei Noah's Ark Lab for kind suggestions and support.

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

## A    FURTHER NOTATIONS

We first demonstrate the definition of Gaussian process.

**Definition 2.** A stochastic process $f(x)$, $x \in \mathbb{R}^d$, is called a Gaussian process if for any finite set of points $x_1, x_2, \ldots, x_n$ in $\mathbb{R}^d$, the random vector $(f(x_1), f(x_2), \ldots, f(x_n))$ follows a multivariate normal distribution. Specifically, a Gaussian process $f$ is completely specified by its mean function $m(x) = \mathbb{E}[f(x)]$ and covariance function $K(x, x') = \text{Cov}(f(x), f(x'))$, and is denoted as:

$$f \sim \mathcal{GP}(m(x), K(x, x')).$$

If $m(x) = 0$, the process is called a centred Gaussian process.

For the notational simplicity, we denote a centred Gaussian process $f \sim \mathcal{GP}(0, K)$ directly by $f \sim K$.

## B    THE STRICTLY POSITIVE DEFINITENESS OF NTK

In this section, we demonstrate the proof of Proposition 1. In the following two subsections, we respectively prove the strictly positive definiteness of the NTK for fully connected network and residual network. For the sake of writing and referencing convenience, we have divided Proposition 1 into Proposition 2 and Proposition 3.

### B.1    FULLY CONNECTED NETWORK

The proof of the strictly positive definiteness of NTK is roughly similar to that in Jacot et al. (2018). However, since our network structure includes several bias terms, which is different from Jacot et al. (2018), we provide the whole proof here to ensure completeness.

**Proposition 2.** *The NTK of fully connected network 2.1 is strictly positive definite on $\mathcal{X}$.*

*Proof of Proposition 2.* Recall that the Hadamard product of positive definite matrices is still positive definite, and the sum of positive definite matrices is also still positive definite. By the recursive form of NTK, the result following from Lemma 4. $\square$

The following lemma is provided to ensure the recursive proof between layers can proceed.

**Lemma 1.** *If kernel function $K_1 : \mathbb{R}^d \times \mathbb{R}^d \to \mathbb{R}$ is strictly positive definite, then the kernel function $K_2$ defined by $K_2(x, x') := \mathbf{E}_{f \sim K_1}[\sigma(f(x))\sigma(f(x'))]$ is also strictly positive definite.*

*Proof.* We prove the lemma by contradiction. Suppose that there exists a set of $\{x_i\}_{i=1,\cdots,n} \subset \mathbb{R}^d$ such that the Gram kernel matrix satisfies that $K_2(X, X)$ is not strictly positive definite. Firstly, it is direct to verify that $K_2(X, X)$ is positive definite. For any $u \in \mathbb{R}^n$, we have

$$u^T K_2(X, X)u = \mathbf{E}_{f \sim K_1}[(u^T \sigma(f(X)))^2] \geq 0. \tag{B.1}$$

Since we suppose that $K_2(X, X)$ is not strictly positive definite, it means that there exists a vector $u_0 \in \mathbb{R}^n$ and $u_0 \neq 0$ such that $u_0^T K_2(X, X)u_0 = 0$. Thus we have

$$u_0^T K_2(X, X)u_0 = \mathbf{E}_{f \sim K_1}[(u_0^T \sigma(f(X)))^2], \tag{B.2}$$

namely, $u_0^T \sigma(f(X)) = 0$ holds almost surely. However, $f(X)$ is a $n$-dimensional Gaussian variable with strictly positive definite Covariance matrix, and thus is non-degenerate. We define $g(v) = \sum_{i=1}^n u_{0,i}\sigma(g(v_i))$, where $u_{0,i}$ is the $i$-th entry of $u_0$. Without loss of generality, we assume that $u_{0,n} \neq 0$. If $u_0^T \sigma(f(X)) = 0$ holds almost surely, we have $g(v) = 0$ almost everywhere with respect to Lebesgue measure. We fix a vector $w \in \mathbb{R}^{n-1}$, and define the new function by $g_n(v_n) = g((w^T, v_n)^T) = \sum_{i=1}^{n-1} u_{0,i}g(w_i) + u_{0,n}g(v_n)$. Since $\sigma$ is non-constant, we have $m_1(A_w) > 0$ where $m_k$ denotes the $k$-dimensional Lebesgue measure and $A = \{v_n | g_n(v_n) \neq 0\}$. Then $m_n(\{v | g(v) > 0\}) = \int_{\mathbb{R}} \int_{\mathbb{R}^{n-1}} \mathbf{1}(v_n \in A_w) \mathrm{d}w \mathrm{d}v_n > 0$ Therefore, we prove the lemma by contradiction. $\square$

**Lemma 2.** *Let $f : [-1, 1] \longrightarrow \mathbb{R}$ be a continuous function with the expansion*

$$f(u) = \sum_{n=0}^{\infty} a_n u^n, \quad u \in [-1, 1], \tag{B.3}$$

*and $k(x, y) = f(\langle x, y \rangle)$ be the dot-product kernel on $\mathbb{S}^d$. Then if $a_n \geq 0$ for all $n \geq 0$ and there are infinitely many $a_n > 0$, then $k$ is strictly positive definite on $\mathbb{S}^d_+ := \{(x_1, x_2, \cdots, x_d) \in \mathbb{S}^d | x_d > 0\}$.*

*Proof.* Let $x_1, \cdots, x_n \in \mathbb{S}^d_+$ be different. The kernel matrix is

$$k(X, X) = \sum_{n=0}^{\infty} a_n M_n, \quad M_n = (\langle x_i, x_j \rangle^n)_{i,j}. \tag{B.4}$$

Since $|\langle x_i, x_j \rangle| < 1$, we have

$$M_n \longrightarrow I_n, \tag{B.5}$$

which is strictly positive definite. In this way, we finish the proof. $\square$

**Lemma 3.** *The coefficients of Maclaurin expansion of $\kappa_0(u), \kappa_1(u)$ are both non-negative and infinitely many terms are positive.*

*Proof.* A direct calculation leads to that

$$\kappa_0(u) = \frac{1}{2} + \frac{1}{\pi} \sum_{n=0}^{\infty} \frac{(2n)!}{4^n (n!)^2 (2n+1)} u^{2n+1}, \tag{B.6}$$

and

$$\kappa_1(u) = \frac{1}{\pi} \left[ u \left( \frac{\pi}{2} + \sum_{n=0}^{\infty} \frac{(2n)!}{4^n (n!)^2 (2n+1)} u^{2n+1} \right) + 1 + \sum_{n=1}^{\infty} \frac{(-1)^{n-1}(2n)!}{4^n (n!)^2 (2n-1)} (-1)^n u^{2n} \right]$$

$$= \frac{1}{\pi} + \frac{u}{2} + \frac{1}{2\pi} \sum_{n=0}^{\infty} \frac{(2n)!}{4^n (n!)(n+1)!} \frac{u^{2n+2}}{2n+1}. \tag{B.7}$$

$\square$

**Lemma 4.** *The kernel function $\Sigma^{(2)}$ is strictly positive definite on $\mathcal{X}$.*

*Proof.* We do a transformation from $\mathbb{R}^d$ to $\mathbb{S}^d_+$ to finish the proof. For $x \in \mathbb{R}^d$, we define $\widetilde{x} := (x, 1)$, which means that add an entry 1 to the end of the vector $x$. Define $\phi : \mathbb{R}^d \to \mathbb{S}^d_+$ by $\phi(x) = \frac{\widetilde{x}}{\|\widetilde{x}\|_2}$.

If we define a kernel function $K_1(x, x') = \langle x, x' \rangle$ on $\mathbb{S}^d_+ \times \mathbb{S}^d_+$ and a kernel function $K_2(x, x') = \mathbf{E}_{f \sim K_1}[\sigma(f(x))\sigma(f(x'))] = \frac{1}{2}\kappa_1(\langle x, x' \rangle)$, then $K_2$ is strictly positive definite on $\mathbb{S}^d_+$ by Lemma 3 and Lemma 2. Actually, we then have that

$$\Sigma^{(2)}(x, x') = 2 \|\widetilde{x}\|_2 K_2(\phi(x), \phi(x')) \|\widetilde{x}\|_2. \tag{B.8}$$

Therefore, we can directly verify that $\Sigma^{(2)}$ is strictly positive definite on $\mathcal{X}$. $\square$

## B.2 RESIDUAL NETWORK

For residual neural networks, since only the first layer in the standard residual network structure includes a bias term, we transform the original kernel into an dot-product kernel on the sphere and then take a Maclaurin expansion to complete the proof.

**Proposition 3.** *The NTK of residual network eq. (2.2) is strictly positive definite on $\mathcal{X}$.*

*Proof.* Note that in Lemma 4, we accomplish the proof through a transformation from $\mathbb{R}^d \to \mathbb{S}^d_+$. We can also define the NTK of ResNet in a similar mannar. It is well known that the sum of a positive definite kernel and a strictly positive definite kernel is still strictly positive definite. Therefore, to derive the strictly positive definiteness of $K^{\mathrm{Res}}$, we only need to consider the strictly positive definiteness of $K_L(\cdot, \cdot)$ on $\mathbb{S}^d_+$, which is shown in Lemma 5.

$\square$

**Lemma 5.** *$K_L(\cdot, \cdot)$ is strictly positive definite on $\mathbb{S}_+^d$ when $L \geq 2$.*

Now we start to prove Lemma 5. We first prove that

**Lemma 6.** *$K_1(\cdot, \cdot)$ is strictly positive definite on $\mathbb{S}_+^d$.*

*Proof.* We have known that $K_1$ is a dot-product kernel. If we denote by $u = x^T x'$. Then we have

$$K_1(x, x') = u + a^2 \kappa_1(u). \tag{B.9}$$

Thus $K_1(\cdot, \cdot)$ is strictly positive definite on $\mathbb{S}_+^d$ through Lemma 2 and Lemma 3. $\qquad \square$

Then one can easily prove Lemma 5 recursively based on the result of Lemma 6.

## C DIVERGENCE OF NNK UNDER INFINITE TIME

### C.1 THE DYNAMIC OF $u$

Before the characteristic of NNK, we first demonstrate the dynamic property of the auxiliary variable defined in 2.5, which is pivotal in our analysis. For $u = (u_1, u_2, \cdots, u_n) \in \mathbb{R}^n$, we have the dynamic equation:

$$
\begin{aligned}
\frac{\mathrm{d}}{\mathrm{d}t} u_i(t) &= \frac{-e^{(2y_i-1)f_t(x_i)}}{\left[1 + e^{(2y_i-1)f_t(x_i)}\right]^2} \cdot (2y_i - 1)\frac{\mathrm{d}}{\mathrm{d}t} f_t(x_i) \\
&= -u_i(1 - u_i)\sum_{j=1}^{n}(2y_i - 1)K_t(x_i, x_j)(2y_j - 1)u_j.
\end{aligned}
\tag{C.1}
$$

If we define the matrix

$$
\begin{aligned}
K^r &:= [(2y_i - 1)K_t(x_i, x_j)(2y_j - 1)]_{j \in [n]}^{i \in [n]} \\
&= \mathrm{diag}(2Y - 1)K_t(X, X)\mathrm{diag}(2Y - 1) \\
&= [K_1^r, K_2^r, \cdots, K_n^r],
\end{aligned}
\tag{C.2}
$$

then we can derive a more explicit dynamic equation

$$\frac{\mathrm{d}}{\mathrm{d}t} u_i(t) = -u_i(1 - u_i)\left[K_i^r\right] u. \tag{C.3}$$

It is worth noting that $K^r$ is also a symmetric matrix as $K_t(X, X)$.

### C.2 THE EXPLICIT FORMULA OF NNK

In this subsection, we introduce the explicit formula of NNK of the fully connected network.

We denote by $D_x^{(l)} = \mathbf{1}(W^{(l)}\alpha^{(l-1)}(x) + b^{(l)} > 0)$, such that we have

$$\alpha^{(l)}(x) = \sqrt{\frac{2}{m_l}} D_x^{(l)}(W^{(l)}\alpha^{(l-1)}(x) + b^{(l)}). \tag{C.4}$$

Then the explicit formula of NNK of multi-layer fully connected network is

$$K_t(x, x') = \sum_{l=1}^{L+1} \langle \nabla_{W^{(l)}} f(x), \nabla_{W^{(l)}} f(x) \rangle + \sum_{l=1}^{L} \langle \nabla_{b^{(l)}} f(x), \nabla_{b^{(l)}} f(x) \rangle \tag{C.5}$$

where the explicit formula is

$$
\sum_{l=1}^{L+1} \langle \nabla_{W^{(l)}} f(x), \nabla_{W^{(l)}} f(x) \rangle
$$

$$
= \sum_{l=1}^{L+1} \left\langle \left[ \prod_{r=l+1}^{L+1} \sqrt{\frac{2}{m_{r-1}}} W^{(r)} D_x^{(r-1)} \right] \alpha^{(l-1)}(x), \left[ \prod_{r=l+1}^{L+1} \sqrt{\frac{2}{m_{r-1}}} W^{(r)} D_{x'}^{(r-1)} \right] \alpha^{(l-1)}(x') \right\rangle
$$

$$
= \sum_{l=1}^{L} \left\langle \left[ \prod_{r=l+1}^{L+1} \sqrt{\frac{2}{m_{r-1}}} W^{(r)} D_x^{(r-1)} \right] \alpha^{(l-1)}(x), \left[ \prod_{r=l+1}^{L+1} \sqrt{\frac{2}{m_{r-1}}} W^{(r)} D_{x'}^{(r-1)} \right] \alpha^{(l-1)}(x') \right\rangle
$$

$$
+ \left\langle \alpha^{(L)}(x), \alpha^{(L)}(x') \right\rangle,
$$

(C.6)

and

$$
\sum_{l=1}^{L} \langle \nabla_{b^{(l)}} f(x), \nabla_{b^{(l)}} f(x) \rangle = \sum_{l=1}^{L} \left\langle \left[ \prod_{r=l+1}^{L+1} \sqrt{\frac{2}{m_{r-1}}} W^{(r)} D_x^{(r-1)} \right], \left[ \prod_{r=l+1}^{L+1} \sqrt{\frac{2}{m_{r-1}}} W^{(r)} D_x^{(r-1)} \right] \right\rangle
$$

(C.7)

## C.3  DIVERGENCE OF NNK UNDER INFINITE TIME

We first introduce new notations. By Proposition 1 , we know the NTK is positive definite and thus the kernel matrix $K(X, X)$ is positive definite for any samples $X$. Denote $\lambda_0 := \lambda_{\min}(K(X, X)) > 0$ to be the minimal eigenvalue of NTK Gram matrix. Also, Let $\widetilde{\lambda}_0(t) := \lambda_{\min}(K_t(X, X))$.

*Proof of Theorem 2.* **Part 1: Fully Connected Network.** We proof the theorem by contradiction. We first assume that there is a kind of parameter initialization such that

$$
\sup_{t \geq 0} \left| K_t(x_i, x_j) - K^{\mathrm{FC}}(x_i, x_j) \right| \leq \frac{\lambda_0}{2n^2},
$$

(C.8)

holds for any $x_i, x_j \in \mathcal{X}$. Proposition 1 ensures that $\lambda_0 > 0$. Through Lemma 7 we have

$$
\inf_{t \geq 0} \widetilde{\lambda}_0(t) \geq \frac{\lambda_0}{2} > 0.
$$

(C.9)

By Lemma 8, we have $(2y_i - 1)f_t(x_i) \to +\infty$ as $t \to \infty$.

Recall that $f_t(x_i) = W_t^{(L+1)} \alpha_t^{(L)}(x_i) = \sum_{j=1}^{m_L} W_{j,t}^{(L+1)} \alpha_{j,t}^{(L)}(x_i)$, where $\alpha_{j,t}^{(L)}(x_i)$ and $W_{j,t}^{(L+1)}(x_i)$ denotes the $j$-th entry of $\alpha_t^{(L)}(x_i)$ and $W_t^{(L+1)}(x_i)$. Therefore, there exists an index $j_0 \in [m_L]$, such that the limit superior of $W_{j_0,t}^{(L+1)} \alpha_{j_0,t}^{(L)}(x_i)$ is infinity, as time $t$ comes to infinity.

We first consider the case that the limit superior of $\alpha_{j_0,t}^{(L)}(x_i)$ is infinity. In equation C.5, directly we can see,

$$
K_t(x_i, x_i) \geq \sum_{l=1}^{L+1} \langle \nabla_{W^{(l)}} f(x), \nabla_{W^{(l)}} f(x) \rangle \geq \langle \alpha^{(L)}(x), \alpha^{(L)}(x) \rangle
$$

$$
= \sum_{j=1}^{m_L} \alpha_{j,t}^{(L)}(x_i) \alpha_{j,t}^{(L)}(x_i) \geq \alpha_{j_0,t}^{(L)}(x_i) \alpha_{j_0,t}^{(L)}(x_i) \to \infty.
$$

(C.10)

It yields the result that $K_t(x_i, x_i)$ diverges to infinity, which contradicts to the assumption in equation C.8.

We then consider the other case that the limit superior of $W_{j_0,t}^{(L+1)}(x_i)\mathbf{1}(\alpha_{j_0,t}^{(L)} > 0)$ is infinity. To work on this, we focus on the bias term in the $L$-th layer, i.e., $b^{(L)}$.

$$
\begin{aligned}
K_t(x,x) &\geq \sum_{l=1}^{L}\langle\nabla_{b^{(l)}}f(x),\nabla_{b^{(l)}}f(x)\rangle \geq \langle\nabla_{b^{(L)}}f(x),\nabla_{b^{(L)}}f(x)\rangle \\
&= \left\langle\left[\sqrt{\frac{2}{m_L}}W^{(L+1)}D_x^{(L)}\right],\left[\sqrt{\frac{2}{m_L}}W^{(L+1)}D_x^{(L)}\right]\right\rangle \\
&\geq \frac{2}{m_L}\left(W_{j_0,t}^{(L+1)}(x_i)\mathbf{1}(\alpha_{j_0,t}^{(L)}>0)\right)^2 \to \infty.
\end{aligned}
\tag{C.11}
$$

It also contradicts to the assumption equation C.8. Therefore, we finish the proof.

**Part 2: Residual Network.** The proof is also accomplished through contradiction. We conduct it in a similar mannar as in the case of fully connected network. We first assume that there is a kind of parameter initialization such that

$$
\sup_{t\geq 0}\left|K_t(x_i,x_j) - K^{\mathrm{Res}}(x_i,x_j)\right| \leq \frac{\lambda_0}{2n^2},
\tag{C.12}
$$

holds for any $x_i, x_j \in \mathcal{X}$. Proposition 1 ensures that $\lambda_0 > 0$. Through Lemma 7 we have

$$
\inf_{t\geq 0}\widetilde{\lambda}_0(t) \geq \frac{\lambda_0}{2} > 0.
\tag{C.13}
$$

By Lemma 8, we have $(2y_i - 1)f_t(x_i) \to +\infty$ as $t \to \infty$. Recall the structure in equation 2.2 of residual network, the network output function $f(x;\theta)$ can be actually decomposed as

$$
f(x;\theta) = W^{(L+1)}\alpha^{(L)}(x);
$$
$$
\alpha^{(L)}(x) = \sqrt{\frac{1}{m_0}}(Ax+b) + \sum_{l=1}^{L}\sqrt{\frac{1}{m_l}}\alpha\left(V^{(l)}\sigma\left(\sqrt{\frac{2}{m_l}}W^{(l)}\alpha^{(l-1)}(x)+b^{(l)}\right)+d^{(l)}\right).
\tag{C.14}
$$

Since for any $x_i$ in training set, we have either $\left\|W_t^{(L+1)}\right\|_2 \to \infty$ or $\left\|\alpha_t^{(L)}(x_i)\right\|_2 \to \infty$ in the sense of the limit superior. Namely, we have either

$$
\limsup_{t\to\infty}\left\|W_t^{(L+1)}\right\|_2 = \infty, \quad \text{or} \quad \limsup_{t\to\infty}\left\|\alpha_t^{(L)}(x_i)\right\|_2 = \infty.
\tag{C.15}
$$

If the former case holds, we have

$$
\left\langle\frac{\partial f_t(x_i)}{\partial d^{(L)}},\frac{\partial f_t(x_i)}{\partial d^{(L)}}\right\rangle = \frac{1}{m_L}\left\|W_t^{(L+1)}\right\|_2^2 \to \infty,
\tag{C.16}
$$

in the sense of the limit superior. And thus the NNK will not converges to the NTK.

If the latter case holds, we also have

$$
\left\langle\frac{\partial f_t(x_i)}{\partial W^{(L+1)}},\frac{\partial f_t(x_i)}{\partial W^{(L+1)}}\right\rangle = \left\|\alpha_t^{(L)}\right\|_2^2 \to \infty,
\tag{C.17}
$$

in the sense of the limit superior.

Therefore, the theorem is proved through contradiction.

$\square$

**Lemma 7.** *Consider fully connected network 2.1 and residual network 2.2. Suppose the NNK uniformly converges to the NTK over the samples, as the network width $m$ comes to infinity, namely,*

$$
\sup_{t\geq 0}\sup_{i,j\in[n]}|K_t(x_i,x_j) - K(x_i,x_j)| \leq \frac{\lambda_0}{2n^2},
\tag{C.18}
$$

*then we can postulate a lower bound of $\widetilde{\lambda}_0$:*

$$
\inf_{t\geq 0}\widetilde{\lambda}_0(t) \geq \frac{\lambda_0}{2}.
\tag{C.19}
$$

*Proof.*

$$|\widetilde{\lambda}_0(t) - \lambda_0| \le \|K_t(X,X) - K(X,X)\|_2 \le \|K_t(X,X) - K(X,X)\|_F$$

$$\le \sum_{i=1}^{n} \sum_{j=1}^{n} |K_t(x_i, x_j) - K(x_i, x_j)| \le \frac{\lambda_0}{2}. \tag{C.20}$$

Thus the lemma is proved. □

**Lemma 8.** *Consider fully connected network 2.1 and residual network 2.2. If we have a positive constant lower bound $C$ of $\widetilde{\lambda}_0(t)$ during training, then the network function will comes to infinity at the sample points $\{x_i\}_{i \in [n]}$, i.e.,*

$$\lim_{t \to +\infty} (2y_i - 1) f_t(x_i) = +\infty. \tag{C.21}$$

*Proof.* Since $K^r = \mathrm{diag}(2Y - 1) K_t(X,X) \mathrm{diag}(2Y - 1)^{-1}$, we know $K^r$ share the same eigenvalues as $K_t(X,X)$. Therefore, we have $\lambda_{\min}(K^r) = \widetilde{\lambda}_0(t)$. Define a function $V(r) = -\sum_{i=1}^{n} \ln(1 - r_i)$, where $r = (r_1, r_2, \cdots, r_n)^T \in [0, 1)^n$. Apparently, we have $V(r) \ge 0$. Given the dynamic of $u$ equation C.3, we can derive the dynamic of $V(u)$:

$$\frac{\mathrm{d}}{\mathrm{d}t} V(u) = \sum_{i=1}^{n} \frac{\partial V}{\partial r_i}\Big|_{r_i = u_i} \frac{\mathrm{d}u_i}{\mathrm{d}t}$$

$$= \sum_{i=1}^{n} \frac{1}{1 - u_i} \frac{\mathrm{d}u_i}{\mathrm{d}t} \tag{C.22}$$

$$= -\sum_{i=1}^{n} u_i [K_i^r]^T u = -u^T K^r u < 0.$$

According to the monotone convergence theorem, we know $V_t = V(u)$ converges as $t \to \infty$. We can also prove that the limit of $V_t$ is exact zero. We give the proof by contradiction.

First, we assume that $\lim_{t \to \infty} V_t = V_* > 0$ holds. Then for any $\varepsilon > 0$, there exists $t_0 > 0$, such that for any $t > t_0$, we have $V_* \le V_t \le V_* + \varepsilon$ holds. We define the following set:

$$\Gamma_\varepsilon = [0, 1 - e^{-(V_* + \varepsilon)}]^n \setminus [0, 1 - e^{\frac{V_*}{n}})^n$$

$$= \{r \in [0, 1)^n \mid 1 - e^{\frac{V_*}{n}} \le \max_{i=1,2,\cdots,n} r_i \le 1 - e^{-(V_* + \varepsilon)}\}. \tag{C.23}$$

We can verify that $V^{-1}[V_*, V_* + \varepsilon] \subset \Gamma_\varepsilon$. Therefore, we have $u \in \Gamma_\varepsilon$ when $t > t_0$. Given that $\Gamma_\varepsilon$ is a compact set, we define $M = \min_{r \in \Gamma_\varepsilon} \|r\|^2 > 0$. When $r \in \Gamma_\varepsilon$, we can derive that

$$r^T K^r r \ge \lambda_{\min}(K^r) \|r\|^2 \ge \lambda_{\min}(K^r) M \ge MC > 0, \tag{C.24}$$

then there is a contradiction that

$$\frac{\mathrm{d}}{\mathrm{d}t} V(u) = -u^T K^r u \le -MC$$

$$\implies V_t \le V_{t_0} + \int_{t_0}^{t} -MC \, \mathrm{d}t \tag{C.25}$$

$$= V_{t_0} - MC(t - t_0) \to -\infty.$$

Thus, we prove that the limit of $V_t$ is zero. Based on the limit of $V_t$, we can also get the convergence result of $u_i$:

$$V(u) \to 0$$

$$\implies 0 < u_i \le -\ln(1 - u_i) \le -\sum_{i=1}^{n} \ln(1 - u_i) = V(u) \to 0. \tag{C.26}$$

Therefore, we have $\lim_{t \to +\infty} (2y_i - 1) f_t(x_i) = +\infty$ as the time $t$ comes to infinity holds for any $i \in [n]$. In this way, we finish the proof. □

**Remark 1.** The result of Lemma 8 can be extended to more general loss functions. Specifically, the influence of the loss function is reflected in equation (C.22). Let $l(x)$ be a convex loss function with $l'(x) < 0$, which appears in training as $l\left((2y_i - 1)f(x_i)\right)$. Under these conditions, the proof of Lemma 8 still holds. We only need to define

$$u := \frac{\partial l\left((2y_i - 1)f(x_i)\right)}{\partial f(x_i)},$$

and set $V(u(\cdot)) = l(\cdot)$ as replacemment. This demonstrates that our proof is general under these assumptions.

