# OpenReview forum: "Divergence of Neural Tangent Kernel in Classification Problems"
_ICLR.cc/2025/Conference — ICLR 2025 Poster_

### Official Review · Reviewer_jqsc · 2024-10-31

**Soundness:** 2
**Presentation:** 3
**Contribution:** 1
**Rating:** 3
**Confidence:** 4

**Summary:**

This paper showed that the multi-layer fully connected neural networks and residual neural networks satisfy the NTK in classification problems. Based on this building, this paper shows the positive definiteness and the convergence of NTK.

**Strengths:**

This paper is easy to follow.

**Weaknesses:**

The main contributions of this paper are trivial since the derivation in Eqs. (2.7)-(3.2) is common-used. Besides, the finding that the empirical NTK does not consistently converge but instead diverges as time approaches infinity has been observed by existing studies.


----

I am very sorry that I failed to upload the complete review comments due to my negligence. I would like to add now:

The main claim of this paper is that the neural network parameters will diverge due to the strict positive definiteness of NTK. This is an interpretation, and the core technology behind it is the estimation of strictly positive definite NTK eigenvalues. This is an existing conclusion [1-2]. Therefore, the technical contribution of this paper is not novel, but more like a reinterpretation.

When people talk about a new interpretation of a formula, they should consider what the new interpretation means. I found that there is a statement, "this finding implies that NTK theory is not applicable in this context, highlighting significant theoretical implications for the study of neural networks in classification problems." Unfortunately, this paper does not provide further analysis.

[1] Q. Nguyen, M. Mondelli, and G. Montufar. Tight bounds on the smallest eigenvalue of the neural tangent kernel for deep relu networks. In Proceedings of the 38th International Conference on Machine Learning, pages 8119–8129, 2021. [2] Shao-Qun Zhang, Zong-Yi Chen, Yong-Ming Tian, ​​Xun Lu. A Unified Kernel for Neural Network Learning. arXiv 2024.

**Questions:**

Nothing.

---

> ### Author Response · Authors · 2024-11-13
>
> There may be some misunderstandings between us, and we would like to take this opportunity to address them one by one.
>
> ---
>
>  ### 1.  ”*The main contributions of this paper are trivial since the derivation in Eqs. (2.7)-(3.2) is common-used.*“
> These equations are not our core results but are used to set up the problem. They are standard in NTK theory. Our theoretical contribution lies in **Theorem 2**.
>
>
> In our paper, the content of Eqs. (2.7)-(3.2) is as follows:
>
> **Equation (2.7):**
> $$
> \begin{aligned}
> \frac{\mathrm{d}}{\mathrm{~d} t} \theta_{t} & =-\nabla_{\theta} \mathcal{L}\left(\theta_{t}\right)=-\sum_{i=1}^{n} \ell^{\prime}\left(\left(2 y_{i}-1\right) f\left(x_{i} ; \theta_{t}\right)\right)\left(2 y_{i}-1\right) \nabla_{\theta} f\left(x_{i} ; \theta_{t}\right)
>  =\sum_{i=1}^{n} \nabla_{\theta} f\left(x_{i} ; \theta_{t}\right)\left(2 y_{i}-1\right) u_{i} .
> \end{aligned}
> $$
>
> **Equation (3.1):**
> $$
> K_{t}\left(x, x^{\prime}\right)=\left\langle\nabla_{\theta} f\left(x ; \theta_{t}\right), \nabla_{\theta} f\left(x^{\prime} ; \theta_{t}\right)\right\rangle .
> $$
>
> **Equation (3.2):**
> $$
> \begin{aligned}
> \frac{\mathrm{d}}{\mathrm{~d} t} f\left(x ; \theta_{t}\right)  =\left[\nabla_{\theta} f\left(x ; \theta_{t}\right)\right]^{T}\left[\frac{\partial \theta_{t}}{\partial t}\right]
>  =\sum_{i=1}^{n}\left[\nabla_{\theta} f\left(x ; \theta_{t}\right)\right]^{T}\left[\nabla_{\theta} f\left(x_{i} ; \theta_{t}\right)\right]\left(2 y_{i}-1\right) u_{i}  =\sum_{i=1}^{n} K_{t}\left(x, x_{i}\right)\left(2 y_{i}-1\right) u_{i} .
> \end{aligned}
> $$
>
> These three equations are not our core results but are used to set up the problem. They are standard in NTK theory. Specifically:
> 1. Equation (2.7) describes the dynamics of neural network parameters under gradient descent.
> 2. Equation (3.1) provides the standard definition of the (empirical) Neural Tangent Kernel (NTK).
> 3. Equation (3.2) describes the dynamics of the neural network function during gradient descent.
>
> Our theoretical contribution lies in **Theorem 2**, which demonstrates the divergence property of the NTK. In Theorem 2, we state that as the training time  $t \to \infty$ , the empirical NTK  $K_t(x, x{\prime})$  may not converges, which offers a perspective that differs from the conventional assumption of NTK convergence in regression tasks. We greatly value your feedback and hope this clarification helps address your concerns.
>
> ---
>
> ### 2. ” *Besides, the finding that the empirical NTK does not consistently converge but instead diverges as time approaches infinity has been observed by existing studies.*“
>
> In fact, to the best of our knowledge, we have not seen studies addressing the divergence of the NTK in previous literature, especially in classification problem. For regression problems under the Mean Squared Error (MSE) loss, it is commonly believed that the NTK converges [Allen-Zhu et al., 2019; Xu and Zhu, 2024].  If you are aware of works that explore the divergence of the NTK, we would be sincerely grateful if you could share them with us. Thank you for your valuable insights.
>
> ---
>
> ### References
>
> [Allen-Zhu et al., 2019] Allen-Zhu, Z., Li, Y., and Song, Z. (2019). "A convergence theory for deep learning via over-parameterization." In _International Conference on Machine Learning_, pages 242–252. PMLR.
> [Xu and Zhu, 2024] Xu, J., and Zhu, H. (2024). "Overparametrized Multi-layer Neural Networks: Uniform Concentration of Neural Tangent Kernel and Convergence of Stochastic Gradient Descent." _Journal of Machine Learning Research_, 25(94), 1–83.

---

> > ### Comment · Reviewer_jqsc · 2024-11-27
> > **response to authors**
> >
> > ----
> >
> > I am very sorry that I failed to upload the complete review comments due to my negligence. I would like to add now:
> >
> > The main claim of this paper is that the neural network parameters will diverge due to the strict positive definiteness of NTK. This is an interpretation, and the core technology behind it is the estimation of strictly positive definite NTK eigenvalues. This is an existing conclusion [1-2]. Therefore, the technical contribution of this paper is not novel, but more like a reinterpretation.
> >
> > When people talk about a new interpretation of a formula, they should consider what the new interpretation means. I found that there is a statement, "this finding implies that NTK theory is not applicable in this context, highlighting significant theoretical implications for the study of neural networks in classification problems." Unfortunately, this paper does not provide further analysis.
> >
> > [1] Q. Nguyen, M. Mondelli, and G. Montufar. Tight bounds on the smallest eigenvalue of the neural tangent kernel for deep relu networks. In Proceedings of the 38th International Conference on Machine Learning, pages 8119–8129, 2021. [2] Shao-Qun Zhang, Zong-Yi Chen, Yong-Ming Tian, ​​Xun Lu. A Unified Kernel for Neural Network Learning. arXiv 2024.

---

> > > ### Author Response · Authors · 2024-11-27
> > >
> > > Thank you for your response; there may be some misunderstandings here.
> > >
> > > Our main contribution is showing the divergence of NNK from the positive definiteness of NTK. This phenomenon is interesting, and the only related results we could find are about the convergence from NNK to NTK. Some papers use MSE loss [Lai et al., 2023; Allen-Zhu et al., 2019] to prove the convergence of NTK; others use the cross-entropy loss function and obtain convergence from NNK to NTK by limiting the training time [Taheri and Thrampoulidis, 2024]. However, under infinite training time with the cross-entropy loss function, the divergence of NTK is first explicitly pointed out by us. The two references you provided explain the positive definiteness of NTK but do not address the divergence of NNK.
> > >
> > > The core of our work lies in deriving the divergence of NNK from the positive definiteness of NTK. Unlike existing work, due to the divergence of NNK, we cannot directly use the convergence from NNK to NTK to approximate the training process. Therefore, we can only construct by contradiction and obtain its divergence from the explicit expression of NNK. This part of the proof is not a reinterpretation of previous work but our new discovery.
> > >
> > > The positive definiteness of NTK has long been well studied [Bietti and Bach, 2020; Arora et al., 2019], including the references you mentioned [Q. Nguyen et al., 2021; Zhang et al., 2024], which we will compare in the main text. We demonstrate positive definiteness in our paper only to ensure the completeness of the proof. After all, slight structural differences (e.g., whether to include bias terms) can lead to entirely different proofs of positive definiteness. For example, for a homogeneous neural network without bias terms, we can transform its NTK into an inner product on the sphere and then expand it to obtain positive definiteness [Bietti and Bach, 2020]; whereas for networks with bias terms, proving positive definiteness may require some recursive methods. But as mentioned earlier, this is not the focus of our work. We will emphasize this point more to prevent potential misunderstandings.
> > >
> > > Regarding the theoretical significance of the divergence of NNK in classification problems, we explain it here. To see the significance of this result, we can compare it with regression problems.  In regression, people can achieve uniform convergence of NNK to NTK, then approximate networks using kernel regression, and thus explicitly obtain properties of generalization ability [Lai et al., 2023; Li et al., 2024a]. However, in classification problems, NNK cannot uniformly converge to NTK. Therefore, during the process of $T \to \infty$, the network can not be approximated by the NTK predictor and what happens to the network may be difficult to analyze from the perspective of NTK. We propose this perspective to inspire future work.
> > >
> > > Thank you for your response. We hope this can address your concerns to some extent. To avoid similar misunderstandings, we will further emphasize our contributions in the article. Thank you for your review! If you have further questions, please feel free to ask us.
> > >
> > > **References**
> > >
> > > - Bietti, A., & Bach, F. (2020). Deep equals shallow for ReLU networks in kernel regimes. *arXiv preprint arXiv:2009.14397*.
> > > - Arora, S., Du, S. S., Hu, W., Li, Z., Salakhutdinov, R. R., & Wang, R. (2019). On exact computation with an infinitely wide neural net. *Advances in Neural Information Processing Systems*, 32.
> > > - Taheri, H., & Thrampoulidis, C. (2024). Generalization and Stability of Interpolating Neural Networks with Minimal Width. *Journal of Machine Learning Research*, 25(82), 1–47.
> > > - Lai, J., Xu, M., Chen, R., & Lin, Q. (2023). Generalization ability of wide neural networks on R. *arXiv preprint arXiv:2302.05933*.
> > > - Li, Y., Yu, Z., Chen, G., & Lin, Q. (2024). On the Eigenvalue Decay Rates of a Class of Neural-Network Related Kernel Functions Defined on General Domains. *Journal of Machine Learning Research*, 25(82), 1–47.
> > > - Allen-Zhu, Zeyuan, Yuanzhi Li, and Zhao Song. (2019.) "A convergence theory for deep learning via over-parameterization." _International conference on machine learning_. PMLR,
> > > - Nguyen, Q., Mondelli, M., & Montufar, G. (2021). Tight bounds on the smallest eigenvalue of the neural tangent kernel for deep ReLU networks. *Proceedings of the 38th International Conference on Machine Learning*, 8119–8129.
> > > - Zhang, S.-Q., Chen, Z.-Y., Tian, Y.-M., & Lu, X. (2024). A Unified Kernel for Neural Network Learning. arXiv preprint.

---

> > > ### Author Response · Authors · 2024-12-03
> > >
> > > Thank you for your valuable feedback
> > >  again. We have carefully addressed your questions. Should you have any further inquiries or require additional references, please do not hesitate to let us know.

---

### Official Review · Reviewer_rp1n · 2024-11-03

**Soundness:** 3
**Presentation:** 3
**Contribution:** 2
**Rating:** 6
**Confidence:** 3

**Summary:**

This paper explores the convergence properties of the Neural Tangent Kernel (NTK) in classification tasks, with a focus on multi-layer fully connected neural networks and residual neural networks. The authors establish that the NTK of both types of networks is strictly positive definite, a key characteristic for comprehending the training behavior of neural networks. Using a contradiction argument, the authors demonstrate that when employing the cross-entropy loss function, the parameters of the neural network tend to diverge due to the strictly positive definiteness of the NTK. The results imply that NTK theory may not hold in the context of classification problems, leading to significant theoretical implications for understanding the dynamics of neural networks.

**Strengths:**

1.	The paper provides valuable insights into the behavior of the NTK in classification problems, highlighting a significant divergence from established understandings in regression tasks.

2.	By demonstrating that NTK does not uniformly converge in classification scenarios, the authors identify a critical limitation of NTK theory.

**Weaknesses:**

1. The claim that current NTK theories might not fully explain generalization properties in classification tasks could be seen as an overgeneralization without comprehensive evidence. A more nuanced discussion on the conditions under which NTK theory might still apply could add depth to the argument.

2. Although the paper recognizes the need for new theoretical tools, it doesn't provide specific directions or approaches for developing these frameworks. Offering some initial ideas or suggestions could help guide future research.

**Questions:**

1. In Theorem 1, it is assumed that the minimum eigenvalue of the NNK matrix has a lower bound. However, the eigenvalues of the NNK Gram matrix approach zero as the number of training examples increases, as demonstrated in the two papers below. Can this observation offer any insights into NTK theory for classification problems?

[1] Lili Su and Pengkun Yang. On learning over-parameterized neural networks: A functional approximation perspective. In Advances in Neural Information Processing Systems, pp. 2637–2646, 2019.

[2] Atsushi Nitanda and Suzuki Taiji. Optimal rates for averaged stochastic gradient descent under neural tangent kernel regime. In International Conference on Learning Representations, 2021.

2. Could the scope be broadened to incorporate other classification loss functions in order to gain a more comprehensive understanding of NTK behavior?

3. What will happen if the data is completely separable or follows a Gaussian mixture in classification problems?

---

> ### Author Response · Authors · 2024-11-27
>
> Thank you for your review. We will now attempt to address your questions.
>
> _**In Theorem 1, it is assumed that the minimum eigenvalue of the NNK matrix has a lower bound. However, the eigenvalues of the NNK Gram matrix approach zero as the number of training examples increases, as demonstrated in the two papers below. Can this observation offer any insights into NTK theory for classification problems?**_
>
> Thank you for pointing out that the eigenvalues of the positive definite NTK indeed decay to zero, and there is no constant lower bound greater than zero. Also, the lower bound $\widetilde{\lambda}_0$ of the eigenvalues of the NNK Gram matrix will also face a similar issue as $n \to \infty$. Regarding this problem, in fact, our "divergence" conclusion was indeed for a fixed $n$, not in the asymptotic sense of $n$. Our conclusion is: For each training with sample size $n$, the NNK diverges during this training process.
>
> To see the significance of this result, we can compare it with regression problems: In regression, regardless of the value of $n$, as long as $m$ is sufficiently large, we can achieve uniform convergence of NNK to NTK, then approximate neural networks using kernel regression, and thus obtain properties of generalization ability [Lai et al., 2023; Li et al., 2024a]. However, in classification problems, NNK cannot uniformly converge to NTK. Although when $n$ is sufficiently large, the lower bound of the gap between NNK and NTK, $\frac{\lambda_0}{2n^2}$, is small, this small lower bound still makes it difficult for us to approximate neural networks using kernel regression.
>
> In summary, what we want to express is "divergence" in a single training, not divergence in the asymptotic sense as $n$ grows. Therefore, although $\frac{\lambda_0}{2n^2}$ is very small, this issue does not affect the main conclusion we want to express. However, we have to admit that it doesn't look that elegant—we just haven't thought of how to weaken this condition yet.
>
> **_Could the scope be broadened to incorporate other classification loss functions in order to gain a more comprehensive understanding of NTK behavior?_**
>
> Yes, our conclusions can indeed be extended to more general loss functions. In fact, the influence of the loss function is reflected in equation (C.22) of our proof. Let the loss function be $l(x)$, which appears in training as $l((2y_i - 1) f(x_i))$. It can be easily verified that if the loss function satisfies: 1) convexity; 2) $l'(x) < 0$, then the proof of equation (C.22) still holds. We only need to let $u \coloneqq \frac{\partial l((2y_i -1) f(x_i))}{\partial f(x_i)}$, and let $V(u(\cdot)) = l(\cdot)$. In the previous version of our paper, we only selected the representative cross-entropy loss function. Thank you for your suggestion; we will add more explanations in the paper to show that our proof is general.
>
> _**What will happen if the data is completely separable or follows a Gaussian mixture in classification problems?**_
>
> Thank you for your suggestion. While we find the question raised by the reviewer interesting, it deviates from the scope of our current work, which does not make assumptions about the data distribution. Developing a comprehensive framework that incorporates such assumptions to weaken previous assumptions (like the positive definiteness of the NTK) and utilizes techniques related to separability [D. Soudry, et al.; H. Taheri and C. Thrampoulidis] is beyond the scope of this paper. We thank you for highlighting this possibility and hope that our study will inspire further research in this direction.
>
> [D. Soudry, et al.] The Implicit Bias of Gradient Descent on Separable Data. ICLR 2018.
>
> [H. Taheri and C. Thrampoulidis] Generalization and Stability of Interpolating Neural Networks with Minimal Width. JMLR 2024.
>
> [Lai et al., 2023] Lai, J., Xu, M., Chen, R., & Lin, Q. (2023). Generalization ability of wide neural networks on . arXiv preprint arXiv:2302.05933.
>
> [Li et al., 2024a] Li, Y., Yu, Z., Chen, G., & Lin, Q. (2024). On the Eigenvalue Decay Rates of a Class of Neural-Network Related Kernel Functions Defined on General Domains. _Journal of Machine Learning Research_, 25(82), 1–47.

---

### Official Review · Reviewer_Nv3G · 2024-11-04

**Soundness:** 2
**Presentation:** 2
**Contribution:** 2
**Rating:** 6
**Confidence:** 3

**Summary:**

This paper investigates the divergent properties of neural network kernels (NNK, empirical NTK) in classification problems using the softmax loss. Specifically, it demonstrates that the NTKs of fully connected neural networks and ResNets are strictly positive, and that model outputs and parameters will diverge as long as the NNK is bounded below by a positive constant during training (Theorem 1 and Corollary 1). Additionally, it is shown that, unlike in regression problems, the NNK is not fixed during training (Theorem 2).

**Strengths:**

- The paper proves the divergence of NNK during training, indicating the inapplicability of NTK theory for regression problems.
- The paper is well-structured and easy to read.

**Weaknesses:**

- The divergence result of the NNK is not particularly surprising, as it is evident that parameters and outputs must diverge to minimize standard classification losses such as logistic, softmax, or exponential losses. For example, refer to the following papers:

[D. Soudry, et al] THE IMPLICIT BIAS OF GRADIENT DESCENT ON SEPARABLE DATA. ICLR 2018.

[Z. Ji and M. Telgarsky] The implicit bias of gradient descent on nonseparable data. COLT, 2019.

- Given the scale of $\lambda_0$, which typically degenerates as $n\to \infty$, I am curious about the significance of the deviation $\lambda_0/2n^2$, which goes to $0$ as well.

- The results do not address the convergence of gradient descent. If the positivity of the NNK (Eq. (5.3)) holds, convergence can be shown as in NTK theory, but this requirement seems redundant for classification problems. The NTK separability assumption, a weaker condition than positivity, is sufficient for proving the convergence of gradient descent. See the following papers for details:

[A. Nitanda, G. Chinot, and T. Suzuki] Gradient descent can learn less over-parameterized two-layer neural networks on classification problems. 2019.

[Z. Ji and M. Telgarsky] Polylogarithmic width suffices for gradient descent to achieve arbitrarily small test error with shallow relu networks. ICLR 2020.

[H. Taheri and C. Thrampoulidis] Generalization and Stability of Interpolating Neural Networks with Minimal Width. JMLR 2024.

- These related works are not discussed in the paper.

- (Minor)
A few typos: (1) Line 110: $f(x_1, f(x_2),$ (2) In the second statement of Theorem 2: Should this be the result for $K_t^{Res}$?

**Questions:**

- The obtained results rely on the positivity of the NNK. Could you clarify the required conditions (e.g., number of neurons) for this assumption?

- Could you discuss the relationships with the related papers mentioned in the weaknesses?

---

> ### Author Response · Authors · 2024-11-19
>
> Thank you for your additional comments. We will now compare our results with the articles you mentioned.
>
> ### The Relationship Between Existing Overfitting Results and This Paper
>
> *"The divergence result of the NNK is not particularly surprising, as it is evident that parameters and outputs must diverge to minimize standard classification losses such as logistic, softmax, or exponential losses."*
>
> Our main difference from this kind of overfitting is that in our setting, we fix $m$ and let $T$ approach infinity. In [Z. Ji and M. Telgarsky] and [H. Taheri and C. Thrampoulidis], the setting is to consider a certain time $T$ and then let $m$ become large enough. In the latter case, we can imagine that $f(x)$ tending to infinity does not directly indicate the divergence of the empirical NTK because $m$ is also changing accordingly.
>
> In fact, this is precisely a subtle difference. To illustrate the distinction between the two more specifically, please allow me to ignore technical details and make a brief argument here:
>
> **Assumption**: For simplicity, we use the assumption that the NTK is positive definite, although, as you said, this is a too strong assumption.
>
> Fix a certain time $T_{\max} \in [0, \infty)$, we can prove that within the time interval $[0, T_{\max}]$, with high probability, the NTK regime holds: $ \parallel W_{t} - W_{0}\parallel_2 \leq O(\sqrt{m})$. Since $T_{\max}$ is finite, we can control the sum of gradients in $[0, T_{\max}]$. Furthermore, we can prove that when $m \to \infty$, the empirical NTK converges uniformly in $[0, T_{\max}]$. Then, as long as we take $T_{\max}$ large enough and combine it with the positive definiteness of the NTK, we can prove that the loss of the neural network becomes arbitrarily small.
>
> It can be seen that the "uniform convergence of NTK" in the above process is different from the "divergence of NTK" in our paper, precisely because of whether we fix $m$ or fix $T$.
>
> So why does our setting fix $m$ first, or in other words, what is the purpose of our paper? As is well known, the fact that the loss of neural networks approaches zero is not the end—people are also concerned about the specific generalization ability of overfitted neural networks, especially what happens during the process of $T \to \infty$ after the neural network has overfitted (e.g., whether so-called benign overfitting occurs). To achieve this, in regression tasks, based on the overfitting of neural networks [Du et al., 2019], people have proved that sufficiently wide neural networks converge uniformly to the corresponding kernel predictor at any time $T \in [0, \infty)$ [Lai et al., 2023; Li et al., 2024a]. Then, they study the properties of the corresponding kernel predictor, thus completely characterizing the generalization ability of over-parameterized neural networks [Li et al., 2024b].
>
> However, in classification tasks, our work aims to show that although networks can still overfit, this is due to a not entirely identical mechanism because over-parameterized neural networks will no longer converge uniformly to the kernel predictor. Therefore, during the process of $T \to \infty$, what happens to the neural network may be difficult to analyze from the perspective of NTK. We propose this perspective to inspire future work.
>
> Thank you very much for the literature references you provided! ! We will compare the above literature in our article and briefly describe the above argument process. This is indeed a subtle but meaningful comparison. Moreover, during the process of writing the rebuttal, we also realized that the theoretical significance in our article is not fully explained; we actually need the above comparison to completely characterize our work. Thank you again for your review comments, which greatly supplement our work.
>
> Here is part of the new paragraph of the literature comparison that we plan to include in the paper, while the discussion of theoretical significance will be included in another section:
>
> > **Recent works, such as [Z. Ji and M. Telgarsky] and [H. Taheri and C. Thrampoulidis], have explored the dynamics of overfitting in neural networks under the regime where the network width $m \to \infty$ and the training time $T$ is fixed and finite. These studies demonstrate that under such conditions, the network will overfit under weaker conditions on NTK. In contrast, our work examines the case where $m$ is fixed and $T \to \infty$. We show that in this setting, the NTK exhibits divergence, preventing uniform convergence to kernel predictors and revealing distinct overfitting dynamics in classification tasks.**

---

> > ### Author Response · Authors · 2024-11-19
> >
> > ### Other Issues
> >
> > - As you mentioned, $\frac{\lambda_0}{2n^2}$ will converge to $0$ as $n$ approaches infinity. We consider that under each $n$, the NTK cannot converge. This issue is somewhat similar to the previous "fixing $m$" or "fixing $T$". Although this does not affect the main conclusion we want to express, we have to admit that it doesn't look that elegant—we just haven't thought of how to weaken this condition yet.
> >
> > - Regarding the positive definiteness of NNK, this only relies on the positive definiteness of NTK and the assumption of uniform convergence of NNK. This proof is reflected in Lemma 7. We apologize for the confusion caused by our wording, and we will make the necessary corrections.
> > - **Typos**: Thank you for pointing out the typos, especially in Theorem 2. We will make corrections.
> >
> > ---
> >
> > **References**
> >
> > [Z. Ji and M. Telgarsky] Polylogarithmic width suffices for gradient descent to achieve arbitrarily small test error with shallow ReLU networks. *ICLR*, 2020.
> >
> > [H. Taheri and C. Thrampoulidis] Generalization and Stability of Interpolating Neural Networks with Minimal Width. *JMLR*, 2024.
> >
> > [Du et al., 2019] Du, S., Lee, J., Li, H., Wang, L., & Zhai, X. (2019). Gradient descent finds global minima of deep neural networks. In *International Conference on Machine Learning* (pp. 1675–1685). PMLR.
> >
> > [Lai et al., 2023] Lai, J., Xu, M., Chen, R., & Lin, Q. (2023). Generalization ability of wide neural networks on $\mathbb{R}$. arXiv preprint arXiv:2302.05933.
> >
> > [Li et al., 2024a] Li, Y., Yu, Z., Chen, G., & Lin, Q. (2024). On the Eigenvalue Decay Rates of a Class of Neural-Network Related Kernel Functions Defined on General Domains. *Journal of Machine Learning Research*, 25(82), 1–47.
> >
> > [Li et al., 2024b] Li, Y., Zhang, H., & Lin, Q. (2024). Kernel interpolation generalizes poorly. *Biometrika*, 111(2), 715–722.

---

> ### Comment · Reviewer_Nv3G · 2024-11-26
>
> Thank you for the detailed response. The paper will be well-positioned in the literature by emphasizing the differences from related works, as the authors responded.
>
> > The fact that the loss of neural networks approaches zero is not the end—people are also concerned about the specific generalization ability of overfitted neural networks
>
> I would note that the work I mentioned in my review comments also proved the generalization of overparameterized models for classification. These results use NTK but do not require the fixed NNK argument. This is why I thought that showing the divergence of NNK is not critical.
>
> While the divergence of NNK for classification is intuitively expected, proving it rigorously is indeed important. I have increased my score to 6 in the expectation that the aforementioned points will be also acknowledged in the revised version.

---

### Official Review · Reviewer_wRPu · 2024-11-04

**Soundness:** 3
**Presentation:** 2
**Contribution:** 4
**Rating:** 8
**Confidence:** 3

**Summary:**

The paper considers the convergence of the neural tangent kernel of fully-connected ReLU networks and ResNets for classification problems trained with cross-entropy loss functions.  The key result of the paper is a proof that, during training (in the gradient-flow approximation), the network parameters diverge due to the strict positive definiteness of the NTK, implying that many results from NTK theory may not be applicable in this context.

**Strengths:**

The result appears to be sound and certainly has relevance insofar as it demonstrates the non-applicability of many NTK results in a common scenario (training with cross-entropy).  The presentation is mostly readable and the derivation appears to be correct, though admittedly I did not dive too deeply into the appendices, and certainly the material appears novel (at least to me).

**Weaknesses:**

While I don't question to novelty of the result in literature, I don't necessarily find it too surprising.  To truly minimize the cross-entropy would require $f(x_i,\theta) \to +\infty$ for $y_i=+1$ (and similarly for the other class).  Without regularization and given infinite time, recalling that the NN is a universal approximator in the infinite-width limit, you should expect precisely this, in which light theorem 1 (and subsequently corollary 1 and theorem 2) is to be expected.  Nevertheless it is important to see this intuition proven formally, so I do not consider this a serious criticism.

Minor points - the presentation of the paper could certainly be improved, for example:

- line 163: you don't optimize the parameters with gradient flow.  Gradient flow is used to (approximately) analyze the optimization process.
- line 185: this is badly written.  You already defined $K_t$ in (3.1) - why are you repeating it in (3.3) as though it was new?
- line 283: "...will converges to NTK with probability as width comes to infinity..." Do you mean "will converge with probability 1" or "will converge in probability"?
- Theorem 2: (4.4) is precisely equivalent to (4.3).  Are these meant to refer to different networks (ie should (4.4) be modified to refer to ResNet)?

**Questions:**

See previous sections.

---

> ### Author Response · Authors · 2024-11-26
>
> Thank you very much for your careful review and recognition of our work. Below, we respond to some of the questions.
>
> __*"While I don't question the novelty of the result in literature, I don't necessarily find it too surprising. To truly minimize the cross-entropy would require $f(x_i,\theta)\to +\infty$ for $y_i=+1$ (and similarly for the other class)."*__
>
> Indeed, some previous works [Z. Ji et al. 2020; H. Taheri et al. 2024] have already proven that for fully connected neural networks under the cross-entropy loss, the loss function approaches zero, which implies that $| f(x_i,\theta)|\to \infty$. However, there is a subtle but important difference between our work and these existing studies. Let me briefly explain, ignoring technical details.
>
> In existing works, the general idea is to first fix a large training duration $T_{\max}$, and then provide a threshold for network width $m_{T_\max}$. Then, letting $T_{\max} \to \infty$ with $m \geq m_{T_\max}$, it can be proven that finally $| f(x_i,\theta)| \to \infty$. However, this kind of overfitting **cannot explain the divergence of NNK**. In fact, we can even prove that under this setting, the NTK remains convergent: Fix a certain time $T_{\max} \in [0, \infty)$, we can prove that within the time interval $[0, T_{\max}]$, with high probability, the NTK regime holds: $\parallel W_t-W_0\parallel_2\leq O(\sqrt{m})$. Since $T_{\max}$ is finite, we can control the sum of gradients in $[0, T_{\max}]$. In this way, we can prove that when $m \to \infty$, the NNK converges uniformly to NTK in $[0, T_{\max}]$. Moreover, based on this, if the NTK is positive definite, then letting $T_{\max} \to \infty$, we can prove that $| f(x_i,\theta)| \to \infty$. In summary, under this setting, the proof of overfitting cannot explain the divergence of NNK, because $m$ also changes with time $T_{\max}$.
>
> Unlike existing works, we aim to conduct a consistent analysis of a neural network, rather than just exploring whether it can overfit. After all, besides overfitting, people are also concerned about the "shape" and generalization property of  network after overfitting (e.g., whether benign overfitting occurs). Therefore, we choose to fix $m$ first, and then let the training time $T$ tend to infinity. This is natural and reasonable for neural networks. Under such a setting, in regression problems, uniform convergence can occur, allowing people to use kernel regression to study the generalization ability of neural networks [Lai et al., 2023; Li et al., 2024a; Li et al., 2024b]. However, in classification problems, according to our results, kernel regression approximation is not possible. That is, the overfitting of neural networks in regression and classification problems is actually due to **different reasons**. This is the theoretical significance revealed by our setting.
>
> This difference in setting makes the analysis of neural network properties more challenging—after all, we cannot directly use the NTK approximation. This is why we devised a rather clever proof by contradiction to prove our conclusion, rather than following existing methods to obtain $|f(x_i,\theta)|\to\infty$, and then directly carry out subsequent proofs based on $|f(x_i,\theta)|\to\infty$.
>
> We indeed overlooked the comparison with existing results in the article, which caused our contributions not to be highlighted. Thank you for your question and suggestion, and we hope this response can alleviate your doubts.
>
> ### Minor points
>
> __*"line 163: you don't optimize the parameters with gradient flow. Gradient flow is used to (approximately) analyze the optimization process."*__
>
> 1. Thank you for your reminder. We indeed did not emphasize this point as you pointed out. In practice, training uses gradient descent, and gradient flow is just an approximation when the step size tends to zero. This approximation has an impact; in fact, many works consider gradient descent and make assumption on the step size (e.g., [Du et al. 2018]). We will use more rigorous expressions here.
>
> __*"line 185: this is badly written. You already defined $K_t$ in (3.1) - why are you repeating it in (3.3) as though it was new?"*__
>
> 2. We  apologize for this typo. Equations (3.1) and (3.3) have the same content. We will delete (3.3), keep only (3.1), and reorganize this paragraph. We did not notice this problem before; thank you for careful reviewing!
>
> __*"line 283: '...will converges to NTK with probability as width comes to infinity...' Do you mean 'will converge with probability 1' or 'will converge in probability'?"*__
>
> 3. What we intended to express is indeed "converges in probability". This is indeed a writing typo and thanks for pointing it out.
>
> __*"Theorem 2: (4.4) is precisely equivalent to (4.3). Are these meant to refer to different networks (i.e., should (4.4) be modified to refer to ResNet)?"*__
>
> 4. Yes, the second part of the theorem is about $K^{\mathrm{Res}}$; this is also our typo. Thank you very much for pointing it out.

---

> > ### Author Response · Authors · 2024-11-26
> > **References**
> >
> > ### References
> >
> > [H. Taheri et al. 2024] Generalization and Stability of Interpolating Neural Networks with Minimal Width. *JMLR*, 2024.
> >
> > [Z. Ji et al. 2020] Polylogarithmic width suffices for gradient descent to achieve arbitrarily small test error with shallow ReLU networks. *ICLR*, 2020.
> >
> > [Lai et al., 2023] Lai, J., Xu, M., Chen, R., & Lin, Q. (2023). Generalization ability of wide neural networks on $\mathbb{R}$. arXiv preprint arXiv:2302.05933.
> >
> > [Li et al., 2024a] Li, Y., Yu, Z., Chen, G., & Lin, Q. (2024). On the Eigenvalue Decay Rates of a Class of Neural-Network Related Kernel Functions Defined on General Domains. *Journal of Machine Learning Research*, 25(82), 1–47.
> >
> > [Li et al., 2024b] Li, Y., Zhang, H., & Lin, Q. (2024). Kernel interpolation generalizes poorly. *Biometrika*, 111(2), 715–722.
> >
> > [Du et al. 2018] Du S, Lee J, Li H, et al. Gradient descent finds global minima of deep neural networks[C] //International conference on machine learning. PMLR, 2019: 1675-1685.

---

> > > ### Comment · Reviewer_wRPu · 2024-12-03
> > > **Response**
> > >
> > > Thanks for clarifying your contribution.  This makes more sense now so I have decided to increase my recommendation to accept.

---

### Meta-Review · Area_Chair_tZnd · 2024-12-24

**Metareview:**

This paper theoretically shows that the trainable parameters of neural network diverges in the NTK regime when the cross-entropy loss is considered. A typical approach of NTK analysis is to show convergence of parameters. However, this paper shows an opposite direction. A numerical experiment supports the theoretical findings.

Although the theoretical novelty would be a bit limited, providing a rigorous proof for the divergence in classification settings is important. This paper gives a nice contribution to this literature. After rebuttals, the reviewers are basically positive on this paper. I recommend acceptance for this paper.

On the other hand,  the following sentence added during the discussion phase "Recent works, such as Ji & Telgarsky (2019); Taheri & Thrampoulidis (2024), have explored the dynamics of overfitting in neural networks under the regime where the network width and the training time is fixed and finite. These studies demonstrate that under such conditions, the network will overfit under weaker conditions on NTK" is confusing because they showed generalization ability of neural network as well as optimization guarantees. I recommend the authors to fix this point.

**Additional Comments On Reviewer Discussion:**

Some reviewers raised concerns on its novelty. However, during the discussion period, the authors responded to this concern and the reviewers admit that there is a novelty. I, the AC, acknowledge this point and recommended acceptance.

---

### Decision · Program_Chairs · 2025-01-22

Accept (Poster)